# Spatiotemporal Variations of Reference Evapotranspiration and Its Climatic Driving Factors in Guangdong, a Humid Subtropical Province of South China

Baoshan Zhao [1,†] , Dongsheng An [1,†], Chengming Yan [1], Haofang Yan [2], Ran Kong [1] and Junbo Su [1,*]

1 South Subtropical Crops Research Institute, Chinese Academy of Tropical Agricultural Sciences, Zhanjiang Experimental and Observation Station for National Long-Term Agricultural Green Development, Zhanjiang 524013, China; zhao_baoshan@outlook.com (B.Z.); dongshengan@catas.cn (D.A.); ycm628@catas.cn (C.Y.); kongran2008@126.com (R.K.)
2 Research Center of Fluid Machinery Engineering and Technology, Jiangsu University, Zhenjiang 212013, China; 1000004265@ujs.edu.cn
\* Correspondence: junbosu@catas.cn
† These authors contributed equally to this work.

**Abstract:** It is of great importance to study the changes in reference evapotranspiration ($ET_0$) and the factors that influence it to ensure sustainable and efficient water resource utilization. Daily $ET_0$ data calculated using the Penman–Monteith method from 37 meteorological stations located within Guangdong Province in the humid zone of southern China from 1960 to 2020 were analyzed. The trend analysis and Mann–Kendall test were used to analyze the time series changes in $ET_0$ and major climatic factors (air temperature ($T$), relative humidity ($RH$), sunshine duration ($SD$), and wind speed ($u_2$)) for over 61 years. Sensitivity and contribution analyses were used to evaluate the driving factors of $ET_0$. The main findings of the study are as follows: (1) the trend in average annual $ET_0$ time series in Guangdong slightly increased at a trend rate of 1.61 mm/10a over the past 61 years, with most stations experiencing an increase in $ET_0$. During the same period, air temperature significantly increased, while $RH$ and $SD$ decreased; $u_2$ also decreased. (2) Sensitivity analysis showed that $ET_0$ was more sensitive to $RH$ and $T$ than $SD$ and $u_2$, with $ET_0$ being most sensitive to $RH$ in spring and winter and $T$ in summer and autumn. (3) The contribution analysis showed that $T$ was the dominant factor for $ET_0$ variation in Guangdong, followed by $SD$. $SD$ was found to be the dominant factor in $ET_0$ changes in areas where the "evaporation paradox" occurred, as well as in spring and summer. The study concludes that the climate in Guangdong became warmer and drier over the past 61 years, and if the current global warming trend continues, it will lead to higher evapotranspiration and drought occurrence in the future.

**Keywords:** climate change; evapotranspiration; meteorological factors; trend analysis; hydrological cycle

## 1. Introduction

Climate change characterized by global warming profoundly impacted agriculture, ecosystems, and human survival and development over the past few decades [1–3]. As a result, changes in the hydrological cycle and its other component processes can be expected worldwide, leading to a series of water resource-related problems [1,4].

Evapotranspiration (*ET*) is one of the most important components of the hydrological cycle and a key parameter in hydrological models and agricultural irrigation management [5,6]. It is a complex process that is not only controlled by climate variables but also influenced by underlying surface conditions, human activities, and other environmental conditions [7]. Therefore, estimating actual *ET* can be challenging [8]. As an alternative method, reference evapotranspiration ($ET_0$), also known as potential evapotranspiration

in hydrometeorology, is used to assess the effects of climate change on evapotranspiration and the hydrological cycle. $ET_0$ is only affected by climatic factors and can reflect the evaporation capacity of the surface atmosphere under specific meteorological conditions in an integrated manner, while excluding the interference of other environmental conditions [9,10].

$ET_0$ is a basic parameter for regional water balance, irrigation scheduling, and water resource management [9,11]. Many past studies investigated the temporal and spatial patterns of $ET_0$, most of which focused on its response to the combined effects of climate change and human activities in different regions [12–19]. Additional studies showed an increasing trend in $ET_0$ in most parts of the world in recent decades, including in the United States [12], China [13,14], India [15,16], and Iran [17]. Some studies reported increasing trends in $ET_0$ (e.g., Ghafouri-Azar et al. [18] in the Korean Peninsula and Liu et al. [19] in the Tibetan Plateau). According to the definition of $ET_0$, the only factors that affect $ET_0$ are climatic variables [9]. Therefore, some studies focused on the relationship between $ET_0$ and influencing climatic variables [20–24]. On a global scale, a decrease in sunshine duration ($SD$) (or solar radiation) was the main cause of evaporation changes at the end of the 20th century [20]. However, due to the non-uniformity of the spatial and temporal distributions of climatic variables, the importance of climatic variables affecting $ET_0$ varies significantly from region to region. Vicente-Serrano et al. [21] analyzed changes in the annual $ET_0$ from 46 stations in Spain and reported increased values of $ET_0$ in the period 1961–2011. They also demonstrated that relative humidity ($RH$), wind speed ($u_2$), and maximum temperature ($T_{max}$) had stronger effects on $ET_0$ than $SD$ and minimum temperature ($T_{min}$). Patle et al. [22] reported that the most sensitive parameter affecting $ET_0$ estimation in the eastern Himalayan region of Sikkim, India, was $T_{max}$, followed by $SD$, whereas $u_2$, $T_{min}$, and $RH$ had a fluctuating effect on mean $ET_0$. Chu et al. [23] studied the effects of climate change on $ET_0$ in Jiangsu, eastern China, finding that $u_2$ contributed the most to $ET_0$, followed by $SD$. Liu et al. [19] reported that changes in $ET_0$ on the Qinghai–Tibet Plateau in the period 1961–2017 mainly depended on air temperature ($T$), followed by $u_2$ and $SD$, whereas $RH$ had a negative effect. There is currently no consensus on the underlying causes of $ET_0$ variation. This issue exists because $ET_0$ is influenced by a combination of changes in climate variables, such as $T$, $RH$, $u_2$, and $SD$, and there is a complex non-linear relationship between $ET_0$ and these parameters, with significant variability existing between these meteorological factors [13,24].

Guangdong, which is located in the south of mainland China, was one of the first regions in China to undergo reform and opening up and has the most outward-looking economy. Due to rapid socio-economic development and human activities, coupled with its proximity to the South China Sea and significant maritime climatic features, both the ocean and the continent have a significant influence on the climate of the region, and changes in $ET_0$ in the region are likely to be complex. In addition, seasonal droughts and urban water shortages were commonly reported in the region in recent years [25,26]. However, there is a lack of study on the spatial and temporal variability and drivers of $ET_0$ in Guangdong.

In this study, the objectives were to analyze the change trends of annual and seasonal $ET_0$ and major climatic factors through collecting meteorological data and calculating $ET_0$ for Guangdong from 1960 to 2020, and to identify the major climatic driving factors of $ET_0$ through quantifying the effects of meteorological variables on $ET_0$. The results of this study can improve our understanding of the factors contributing to $ET_0$ changes and the impact of climate change on the hydrological cycle in Guangdong Province. It is anticipated that the outcomes of this study will improve guidance for agricultural production and economic development in this vitally important region.

## 2. Materials and Methods

### 2.1. Study Area and Meteorological Data

This study was conducted in Guangdong Province (20°09′ N~25°31′ N, 109°45′ E~117°20′ E), which is located in the humid south of China and has a tropical and subtropical monsoon

climate; the province covers an area of approximately 179,700 km$^2$. The region has abundant sunshine, heat, and water resources, with an annual mean temperature of 22.3 °C; annual sunshine duration of 1745.8 h; and annual precipitation of 1789.3 mm, which varies between 1300 and 2500 mm. However, the region has unevenly distributed water resources, with frequent floods in summer and autumn, and droughts in winter and spring. In addition, the region experiences water shortages caused by population growth, climate change, and water pollution [27]. Understanding climate change and its impact on the hydrological cycle is crucial, as the area has strong evapotranspiration (more than half of the total precipitation) [28].

This study used daily meteorological data from 37 stations (Figure 1) located in Guangdong, including average temperature ($T_{mean}$, °C), maximum temperature ($T_{max}$, °C), minimum temperature ($T_{min}$, °C), relative humidity (*RH*, %), wind speed at 10m height ($u_{10}$, m/s), and sunshine duration (*SD*, h); these data were obtained from the China Meteorological Administration (CMA). The analysis period ranged from 1960 to 2020. Table 1 provides basic characteristics of the meteorological stations, such as latitude, longitude, and altitude. The FAO Penman–Monteith equation was used to calculate the daily $ET_0$ for each station. Some years were excluded from the analysis due to the missing data from several stations, including Station Zhuhai (years 1960 to 1964), Station Fengshun (years 2016 to 2020), and Station Jiexi (years 1965 to 1969). Furthermore, routine quality checks and error correction were performed on the meteorological data according to the methodology of Peterson et al. [29]. The four seasons were divided into spring (March–May), summer (June–August), autumn (September–November), and winter (December–February of the following year).

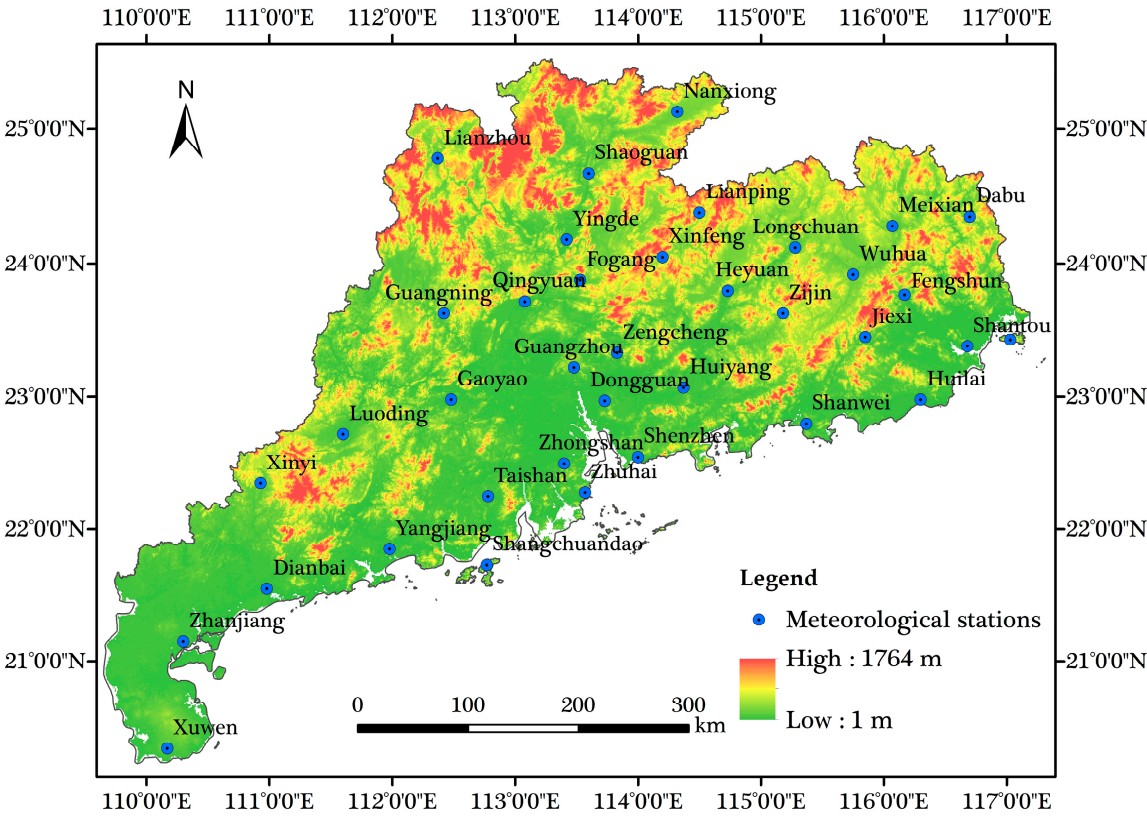

**Figure 1.** Distribution of meteorological stations and altitude map in Guangdong.

**Table 1.** Basic information of meteorological stations used in study area.

| Station Name | Station Code | Latitude (°) | Longitude (°) | Elevation (m) | T (°C) | P (mm year$^{-1}$) | ET$_0$ (mm year$^{-1}$) |
|---|---|---|---|---|---|---|---|
| Nanxiong | 57996 | 25.08 | 114.25 | 149.7 | 20.60 | 1516.93 | 1086.92 |
| Lianzhou | 59072 | 24.82 | 112.37 | 131.7 | 20.59 | 1630.72 | 1017.50 |
| Shaoguan | 59082 | 24.67 | 113.60 | 121.3 | 21.24 | 1598.76 | 1102.62 |
| Fogang | 59087 | 23.88 | 113.52 | 97.2 | 21.87 | 2185.18 | 1110.86 |
| Yingde | 59088 | 24.18 | 113.42 | 74.4 | 21.86 | 1875.78 | 1125.04 |
| Lianping | 59096 | 24.38 | 114.50 | 283.9 | 21.06 | 1761.46 | 1045.66 |
| Xinfeng | 59097 | 24.03 | 114.22 | 269.3 | 21.29 | 1902.73 | 1021.50 |
| Longchuan | 59107 | 24.12 | 115.28 | 179.6 | 21.75 | 1676.19 | 1113.47 |
| Dabu | 59116 | 24.35 | 116.70 | 80.1 | 22.29 | 1504.85 | 1063.31 |
| Meixian | 59117 | 24.28 | 116.07 | 116.0 | 22.34 | 1497.09 | 1125.14 |
| Guangning | 59271 | 23.63 | 112.42 | 92.7 | 22.05 | 1721.33 | 1048.16 |
| Gaoyao | 59278 | 22.98 | 112.48 | 60.0 | 22.96 | 1641.14 | 1132.58 |
| Qingyuan | 59280 | 23.72 | 113.08 | 79.2 | 22.48 | 2125.33 | 1150.73 |
| Guangzhou | 59287 | 23.22 | 113.48 | 70.7 | 22.81 | 1809.36 | 1123.41 |
| Dongguan | 59289 | 22.97 | 113.73 | 56.0 | 23.14 | 1826.26 | 1195.20 |
| Heyuan | 59293 | 23.80 | 114.73 | 71.1 | 22.41 | 1922.63 | 1154.25 |
| Zengcheng | 59294 | 23.33 | 113.83 | 30.7 | 22.67 | 1965.49 | 1155.53 |
| Huiyang | 59298 | 23.07 | 114.37 | 108.5 | 22.87 | 1751.19 | 1191.59 |
| Wuhua | 59303 | 23.92 | 115.75 | 135.9 | 22.18 | 1499.91 | 1151.35 |
| Zijin | 59304 | 23.63 | 115.18 | 176.6 | 21.86 | 1706.80 | 1066.10 |
| Jiexi | 59306 | 23.45 | 115.85 | 80.9 | 22.42 | 2063.74 | 1123.96 |
| Fengshun | 59310 | 23.77 | 116.18 | 45.3 | 22.43 | 1830.54 | 1147.24 |
| Shantou | 59316 | 23.38 | 116.68 | 2.3 | 22.39 | 1556.30 | 1186.02 |
| Huilai | 59317 | 22.98 | 116.30 | 42.0 | 22.68 | 1792.25 | 1210.90 |
| Nan'ao | 59324 | 23.43 | 117.03 | 8.0 | 22.13 | 1357.39 | 1284.52 |
| Xinyi | 59456 | 22.35 | 110.93 | 141.4 | 23.47 | 1790.40 | 1201.36 |
| Luoding | 59462 | 22.72 | 111.60 | 60.0 | 23.12 | 1373.51 | 1114.67 |
| Taishan | 59478 | 22.25 | 112.78 | 33.1 | 22.89 | 1965.12 | 1183.24 |
| Zhongshan | 59485 | 22.50 | 113.40 | 33.7 | 22.97 | 1859.13 | 1134.09 |
| Zhuhai | 59488 | 22.28 | 113.57 | 51.4 | 23.17 | 2031.49 | 1271.63 |
| Shenzhen | 59493 | 22.55 | 114.00 | 63.0 | 23.36 | 1911.09 | 1252.23 |
| Shanwei | 59501 | 22.80 | 115.37 | 16.7 | 22.85 | 1899.26 | 1229.55 |
| Zhanjiang | 59658 | 21.15 | 110.30 | 53.4 | 23.84 | 1675.38 | 1214.12 |
| Yangjiang | 59663 | 21.85 | 111.98 | 90.3 | 23.11 | 2353.95 | 1189.67 |
| Dianbai | 59664 | 21.55 | 110.98 | 31.8 | 23.82 | 1550.61 | 1220.74 |
| Shangchuan Island | 59673 | 21.73 | 112.77 | 21.9 | 23.20 | 2244.92 | 1271.17 |
| Xuwen | 59754 | 20.25 | 110.17 | 11.4 | 24.54 | 1393.28 | 1271.18 |

### 2.2. Reference Evapotranspiration Computation

As measured $ET_0$ values were unavailable, we calculated daily $ET_0$ using the FAO56 Penman-Monteith (PM) method, which is the most widely used and accurate method for estimating $ET_0$ across various climatic regions. The equation is expressed as follows [9]:

$$ET_0 = \frac{0.408\Delta(R_n-G) + \gamma\frac{900}{T+273}u_2(e_s - e_a)}{\Delta+\gamma(1 + 0.34u_2)} \tag{1}$$

where $R_n$ is the net radiation (MJ m$^{-2}$ d$^{-1}$), $G$ is the soil heat flux (MJ m$^{-2}$ d$^{-1}$), $T$ is the mean daily air temperature at 2 m height (°C), $e_s$ is the saturation vapor pressure (kPa), $e_a$ is the actual vapor pressure (kPa), $\Delta$ is the slope of the vapor pressure curve (kPa °C$^{-1}$), and $\gamma$ is the psychrometric constant (kPa °C$^{-1}$).

To convert the wind speed data observed at the meteorological station, which were measured at 10 m above ground level, to the corresponding value at 2 m height, we used Equation (2) [9]:

$$u_2 = u_z \frac{4.87}{ln(68.7z - 5.42)} \tag{2}$$

where $u_z$ is the wind speed at a height of $z$ m above ground level (m s$^{-1}$).

### 2.3. Climatic Trend

Climate tendency refers to the changing trend of meteorological variables over time, which can be estimated using an ordinary linear regression equation, as given using the following equation [30,31]:

$$y(t) = at + b \tag{3}$$

where $t$ represents the long-time series (year), $y(t)$ is the $ET_0$ and other meteorological variables corresponding to $t$, $a$ is the linear slope, and $b$ is the intercept. In general, the climatic tendency rate ($\beta$) is equal to $10a$ with a unit of value per decade [30].

The significance of the trends in climatic series is evaluated using the Mann–Kendall trend test technique (MK test). The MK test is a rank-based non-parametric method that is widely applied for trend detection in hydro-climatic time series [32,33]. The MK test is described as follows:

$$S = \sum_{i=1}^{n-1} \sum_{j=i+1}^{n} \text{sgn}(x_j - x_i) \tag{4}$$

where

$$\text{sgn}(x_j - x_i) = \begin{cases} +1 & (x_j > x_i) \\ 0 & (x_j = x_i) \\ -1 & (x_j < x_i) \end{cases} \tag{5}$$

where $x_i$ and $x_j$ are the sequential data values, and $n$ is the length of the data set. The mean and variance of the statistic $S$ are given as:

$$E(S) = 0 \tag{6}$$

$$V(S) = \left[ n(n-1)(2n+5) - \sum_{i=1}^{n} t_i i(i-1)(2i+5) \right] \tag{7}$$

where $t$ is the extent of any given time.

The standardized statistic $Z$ for a one-tailed test is formulated as follows:

$$Z = \begin{cases} (S-1)/\sqrt{\text{var}(S)} & (S > 0) \\ 0 & (S = 0) \\ (S+1)/\sqrt{\text{var}(S)} & (S < 0) \end{cases} \tag{8}$$

where a positive value of $Z$ denotes an increasing trend, and a negative value indicates a decreasing trend. $|Z| > 1.96$ and $2.32$ indicate passing the significance level test of 0.05 and 0.01, respectively.

### 2.4. Assessing the Impact of Climate Variables on $ET_0$

In this study, the impact of meteorological factors on $ET_0$ was assessed through combining the sensitivity analysis with the contribution rate of a single climate factor to $ET_0$.

Sensitivity analysis is a widely used method to identify the changes in the dependent variable ($ET_0$) caused by the change in an independent meteorological variable [13,14,32], and the sensitivity coefficient is defined by [34]:

$$S_x = \lim_{\Delta x \to 0} \frac{\Delta ET_0 / ET_0}{\Delta x / x} = \frac{\partial ET_0}{\partial x} \cdot \frac{x}{ET_0} \tag{9}$$

where $\Delta x$ is the relative change in the model input value $x$, $x$ is the meteorological factors, $\Delta ET_0$ is the relative change in $ET_0$ induced by $\Delta x$, and $S_x$ is the dimensionless sensitivity coefficient. A positive (negative) $S_x$ means that $ET_0$ increases (decreases) with the increase in meteorological factors. Larger $|S_x|$ means higher sensitivity of $ET_0$ to meteorological factors. In order to quantitatively assess the sensitivity of $ET_0$ to different meteorological factors, the $S_x$ was divided into four levels, as shown in Table 2 [35].

**Table 2.** Sensitivity coefficient level classification.

| $|S_x|$ | Sensitivity Level |
|---|---|
| $|S_x| < 0.05$ | Negligible |
| $0.05 \leq |S_x| < 0.20$ | Moderate |
| $0.20 \leq |S_x| < 1.00$ | High |
| $|S_x| \geq 1.00$ | Very high |

Sensitivity analysis cannot determine the actual contribution of each variable change to $ET_0$. In order to quantify the contributions of meteorological variables to the change trend in $ET_0$, we calculated the contribution rate ($C_x$) through multiplying the multi-year relative change rate of meteorological factors by its sensitivity coefficient, as shown in the following equation [36,37]:

$$C_x = S_x \cdot R_c \tag{10}$$

$$R_c = \frac{n \cdot a}{\overline{x}} \tag{11}$$

where $R_c$ is the relative change rate of certain meteorological factors and $ET_0$ (%); $a$ is the linear slope, as mentioned in Equation (3); and $\overline{x}$ is the mean of the meteorological factor time series.

In this study, ArcGIS 10.8 was used to map the distribution of meteorological stations and spatial variation in $ET_0$ in the study area, Matlab 2018a was used for MK testing, IBM SPSS Statistics 26 was used for significance analysis, and Origin 2021 was used for other plots.

## 3. Results

### 3.1. Change Trends of Climatic Factors

The variations in climatic factors, which were averaged based on the 37 meteorological stations in Guangdong from 1960 to 2020, are shown in Figure 2 and Table 3. In line with the global warming trend, the *T* in the region exhibited a significant increase ($p < 0.01$) with a climate tendency rate of 0.19 °C/10a. The average annual *RH* was 78.51%, which declined significantly at a rate of $-0.42\%$/10a, indicating a trend of drought in the atmosphere, with the most significant reduction occurring around 2010. Sunshine, which directly reflects solar radiation, is the energy source driving changes in other factors, such as *T*, *RH*, $u_2$, and $ET_0$. *SD* in Guangdong ranged from 4.26 to 6.34 h, with a multi-year average of 4.99 h, and showed a significant decreasing trend ($p < 0.01$) with a climate trend rate of $-0.10$ h/10a (Figure 2c). The decline in *SD* could be related to human activities and urbanization, which cause air pollution and an increase in aerosols in the air. In contrast, the multi-year average value of $u_2$ was 1.56 m/s, with a variation range of 1.39 m/s to 1.74 m/s, and showed a non-significant decreasing trend in spring, autumn, and winter (Figure 2d).

On the seasonal scale, *T* displayed a significant declining trend ($p < 0.01$) in all four seasons, and the increasing trend was stronger in cooler seasons than in warmer seasons. Autumn and winter warming rates were 0.24 °C/10a and 0.26 °C/10a, respectively, which were higher than the respective spring and summer warming rates of 0.15 °C/10a and 0.16 °C/10a. *RH* showed a decreasing trend in all seasons, with significant decreases in summer ($p < 0.01$), spring, and autumn ($p < 0.05$). Similarly, *SD* showed a decreasing trend in all seasons, with significant decreases in summer and winter ($p < 0.05$). $u_2$ showed a non-significant decreasing trend in spring, autumn, and winter and a significant increasing trend in summer ($p < 0.01$), with a tendency rate of 0.02 m/(s·10a).

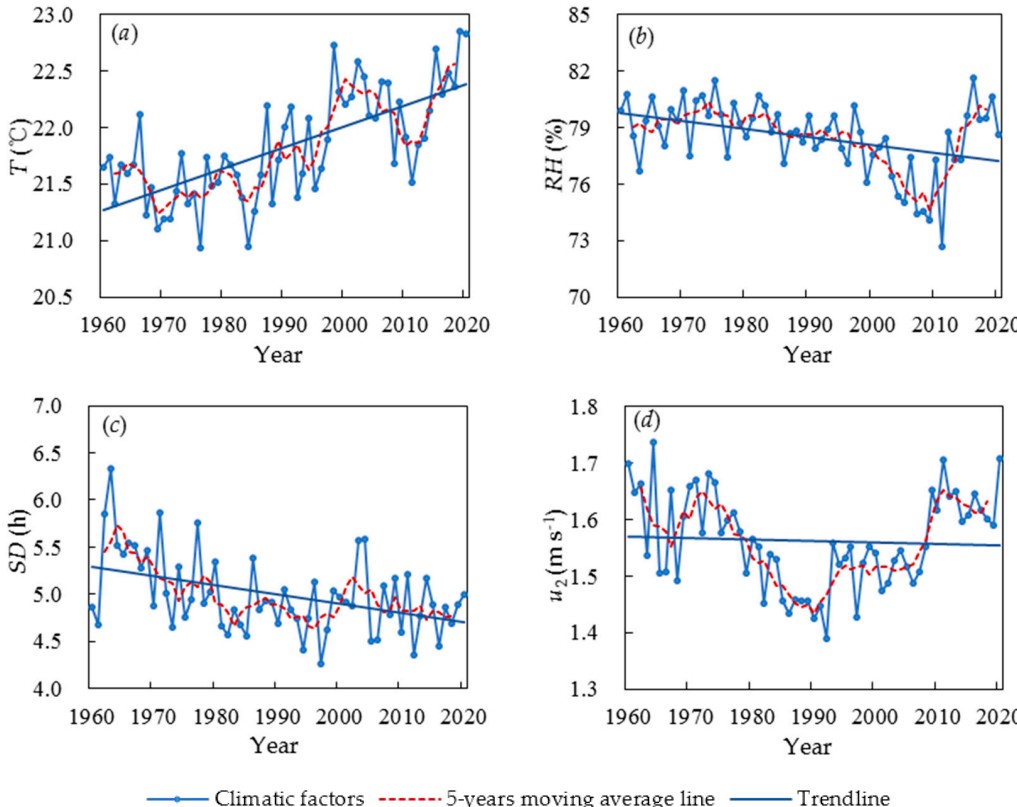

**Figure 2.** Annual variations and linear trends of meteorological variables in Guangdong in period 1960–2020: (**a**) temperature (*T*), (**b**) relative humidity (*RH*), (**c**) sunshine duration (*SD*), and (**d**) wind speed ($u_2$).

**Table 3.** Trend analysis results of climatic variable with linear regression, MK test.

| Climatic Factors | Mean Value | Linear Regression | | MK Test | | |
|---|---|---|---|---|---|---|
| | | Slope | *Std* | Z | *p*-Value | Change Point (Year) |
| Spring | | | | | | |
| *T* (°C) | 22.25 | 0.015 | 0.74 | 2.41 | 0.008 | 2000, 2011 |
| *RH* (%) | 82.20 | −0.038 | 2.51 | −2.38 | 0.046 | 1963, 1994 |
| *SD* (h) | 3.50 | −0.008 | 0.74 | −0.94 | 0.132 | 1968 |
| $u_2$ (m s$^{-1}$) | 1.53 | −0.001 | 0.09 | −1.60 | 0.083 | 1964 |
| Summer | | | | | | |
| *T* (°C) | 28.65 | 0.016 | 0.30 | 5.63 | 0.000 | 1993 |
| *RH* (%) | 82.03 | −0.037 | 1.61 | −3.30 | 0.003 | 1983 |
| *SD* (h) | 6.28 | −0.010 | 0.60 | −2.30 | 0.027 | 1979 |
| $u_2$ (m s$^{-1}$) | 1.45 | 0.002 | 0.08 | 3.18 | 0.004 | 2010 |
| Autumn | | | | | | |
| *T* (°C) | 23.99 | 0.024 | 0.50 | 5.30 | 0.000 | 1995 |
| *RH* (%) | 75.27 | −0.057 | 3.04 | −2.47 | 0.014 | 1966 |
| *SD* (h) | 6.01 | -0.009 | 0.69 | −1.86 | 0.086 | 2011 |
| $u_2$ (m s$^{-1}$) | 1.58 | −0.0006 | 0.12 | −0.29 | 0.458 | - |
| Winter | | | | | | |
| *T* (°C) | 15.01 | 0.026 | 0.94 | 3.58 | 0.000 | 1991, 2011 |
| *RH* (%) | 74.42 | −0.036 | 3.31 | −1.60 | 0.142 | 1997 |
| *SD* (h) | 4.17 | −0.012 | 0.74 | −1.30 | 0.032 | - |
| $u_2$ (m s$^{-1}$) | 1.69 | −0.001 | 0.13 | −0.22 | 0.276 | - |

**Table 3.** *Cont.*

| Climatic Factors | Mean Value | Linear Regression | | | MK Test | | |
| --- | --- | --- | --- | --- | --- | --- | --- |
| | | Slope | *Std* | *Z* | *p*-Value | Change Point (Year) |
| Annual | | | | | | |
| $T$ (°C) | 22.47 | 0.019 | 0.35 | 5.41 | 0.000 | 1999 |
| $RH$ (%) | 78.48 | −0.042 | 1.73 | −3.33 | 0.002 | 1987 |
| $SD$ (h) | 4.99 | −0.010 | 0.37 | −2.73 | 0.001 | 1973 |
| $u_2$ (m s$^{-1}$) | 1.56 | −0.0002 | 0.08 | −0.38 | 0.702 | - |

Note: $T$ is temperature, $RH$ is relative humidity, $SD$ is sunshine duration, $u_2$ is wind speed, slope is trend based on linear regression, *Std* is standard deviation of linear regression, and $Z$ is Mann–Kendall test statistic.

### 3.2. Spatial and Temporal Variation Characteristics of $ET_0$

Due to the non-uniformity of the distribution of climatic factors, the spatial and temporal distribution of $ET_0$ in the study area is uneven. Figure 3a shows the spatial distribution and trend of annual average $ET_0$ in the study area, which shows that $ET_0$ gradually increased from north to south in spatial distribution. The differences in $ET_0$ among meteorological stations were evident, with higher $ET_0$ values found in coastal areas, such as western Guangdong and the Pearl River Delta (PRD). The three stations with the highest $ET_0$ values were Nan'ao (1284.52 mm), Zhuhai (1271.63 mm), and Xuwen (1271.18 mm), while the lowest three stations were Lianzhou (1017.50 mm), Xinfeng (1021.50 mm), and Lianping (1045.66 mm).

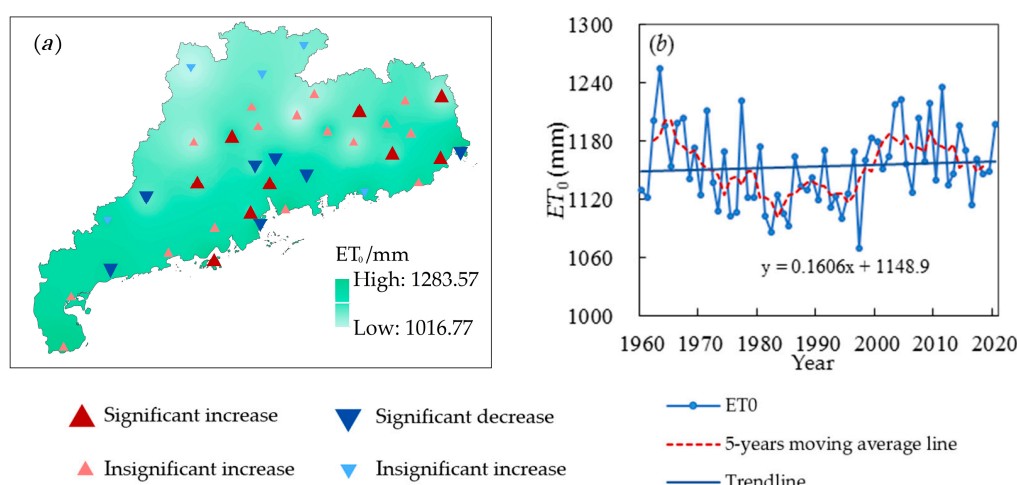

**Figure 3.** Spatial distribution and trend of annual $ET_0$ in study area. (**a**) Spatial variations, (**b**) Temporal variations.

Of the 37 meteorological stations in the study area, 25 stations showed an increasing trend in $ET_0$, of which 9 stations showed a increasing trend with a selected significance level of 0.05. The stations with significant increases in $ET_0$ were found in Qingyuan, Gaoyao, Dongguan, Zhongshan, and Shangchuan Island in the PRD, as well as in Dabu, Jiexi, Shantou, and Longchuan in the east. On the other hand, 12 stations showed a decreasing trend in $ET_0$, with 7 stations showing a significant decrease, including Luoding and Dianbai in the west; Guangzhou, Zengcheng, Zhuhai, and Huiyang in the PRD; and Nan'ao in the east.

In terms of interannual variations, the overall annual $ET_0$ in the study area showed a slightly increasing trend (Figure 3b), with a climatic tendency rate of 1.61 mm/10a and an insignificant decreasing trend (Table 4). The interannual $ET_0$ was unevenly distributed, with a variation range between 1069.27 and 1254.56 mm and a multi-year average of 1142.45 mm.

**Table 4.** Temporal trend analysis of seasonal $ET_0$ with linear regression and MK analysis.

| Season | Mean Value | Linear Regression | | MK Test | | |
|---|---|---|---|---|---|---|
| | | Slope | *Std* | *Z* | *p*-Value | Change Point (Year) |
| Spring | 3.00 | −0.0239 | 22.29 | 0.24 | 0.885 | 2000 |
| Summer | 4.32 | 0.0013 | 18.83 | 0.03 | 0.993 | 2003 |
| Autumn | 3.29 | 0.1004 | 15.92 | 0.91 | 0.397 | 1990 |
| Winter | 2.01 | 0.0828 | 12.97 | 1.23 | 0.392 | 2002 |
| Annual | 3.15 | 0.1606 | 40.58 | 0.83 | 0.594 | 2003 |

Note: Slope is trend based on linear regression, *Std* is standard deviation, and *Z* is Mann–Kendall test statistic.

The spatial and temporal distribution of $ET_0$ in the study area varies across different seasons. Figure 4a–d show the spatial distribution characteristics and trends of multi-year average $ET_0$ at each station during the four seasons. During spring, the variation range of $ET_0$ is between 241.25 and 347.86 mm, and the highest $ET_0$ values are observed in Xuwen. Among the 37 stations in the study area, 23 stations show an increasing trend in $ET_0$, with 4 stations, such as Shantou and Dongguan, showing a significant increase; only 1 of the remaining 14 stations showing a significant decrease. During summer, $ET_0$ variation range is 355.14–431.06 mm, with high $ET_0$ values occurring in regions such as Zhanjiang and Zhuhai. Within the study area, 20 stations have an increasing trend in $ET_0$, while 17 stations have a decreasing trend. Among them, seven stations show a significant increase, while five stations show a significant decrease. The variation ranges of $ET_0$ in autumn and winter are 252.20 to 358.41 mm and 133.40 to 223.20 mm, respectively. The high $ET_0$ values are mainly concentrated in coastal areas, such as Zhuhai, Shenzhen, and Shanwei. In autumn, 24 stations show an increasing trend in $ET_0$, while 13 stations show a decreasing trend, with 8 stations showing a significant increase and 5 stations showing a significant decrease. During winter, $ET_0$ at 26 stations show an increasing trend, with 8 stations showing a significant increase; however, 11 stations show a decreasing trend, and 2 stations showing a significant decrease. The trend in $ET_0$ changes in different seasons also show that the study area has the most stations with increasing $ET_0$ in winter, while there are relatively more stations with decreasing $ET_0$ trends in summer.

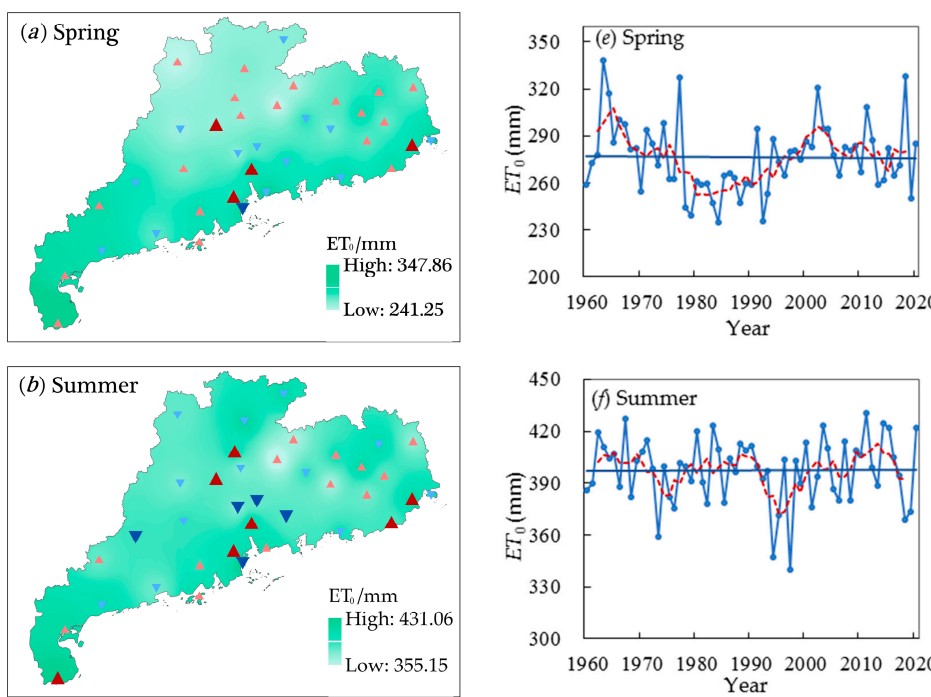

**Figure 4.** *Cont.*

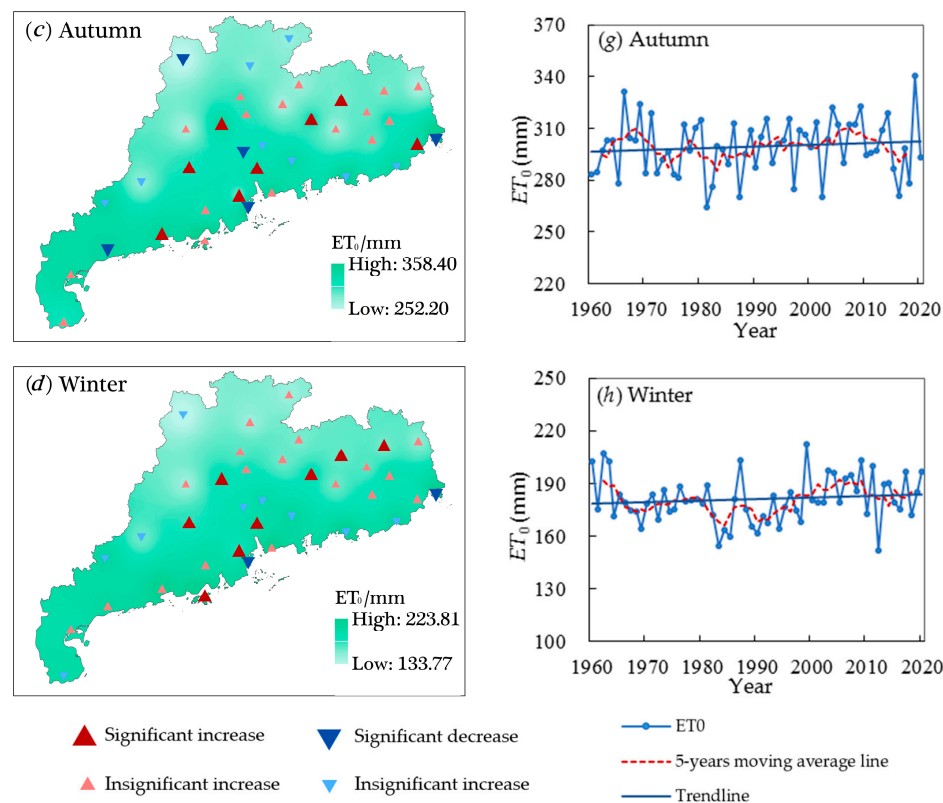

**Figure 4.** Spatial distribution (**a**–**d**) and trend (**e**–**h**) of seasonal $ET_0$ in study area.

The annual $ET_0$ tends to increase in all seasons except spring, albeit not significantly. The climatic tendency rate of $ET_0$ is highest in autumn and winter, being 1.00 mm/10a and 0.83 mm/10a, respectively, and lowest in spring and summer, being −0.24 mm/10a and 0.01 mm/10a, respectively. $ET_0$ is unevenly distributed throughout the year in the study area, with the highest $ET_0$ in summer (34.5% of the year), followed by autumn (299.27 mm), which accounts for 25.9% of the year, and spring (276.10 mm), which accounts for 23.9% of the year. However, in some years, spring $ET_0$ is higher than autumn. Winter $ET_0$ is the smallest (80.87 mm), accounting for only 15.7% of the year.

### 3.3. Sensitivity Analysis of $ET_0$ to Climatic Factors

The spatial distributions of the sensitivity coefficients ($S_x$) of annual $ET_0$ for each climatic factor were analyzed and visualized in Figure 5. The $S_x$ of $ET_0$ to $T$ ranged from 0.49 to 0.75, with an average of 0.65, as well as an increasing trend from north to south. Moreover, the $S_x$ values were higher in the southern coastal areas, with most regions having values greater than 0.70. The $S_x$ of $ET_0$ to $RH$ ranged from −1.53 to −0.36, with an average value of −0.74. The spatial distribution of $|S_x|$ showed an increasing trend from north to south, with $|S_x|$ in the southern coastal areas having values greater than 1.00. However, the spatial distribution difference of the Sx of $ET_0$ to $SD$ and $u_2$ was not significant, ranging from 0.23 to 0.30 and 0.06 to 0.13, with an average of 0.26 and 0.09, respectively. The sensitivity coefficients of $ET_0$ were positive for $T$, $SD$, and $u_2$ and negative for $RH$. Therefore, this result indicates that $ET_0$ in the study area increases with $T$, $SD$, and $u_2$ and decreases with $RH$. The analysis of the sensitivity of $ET_0$ to climatic factors showed that $ET_0$ was highly sensitive to $T$, $RH$, and $SD$ and moderately sensitive to $u_2$.

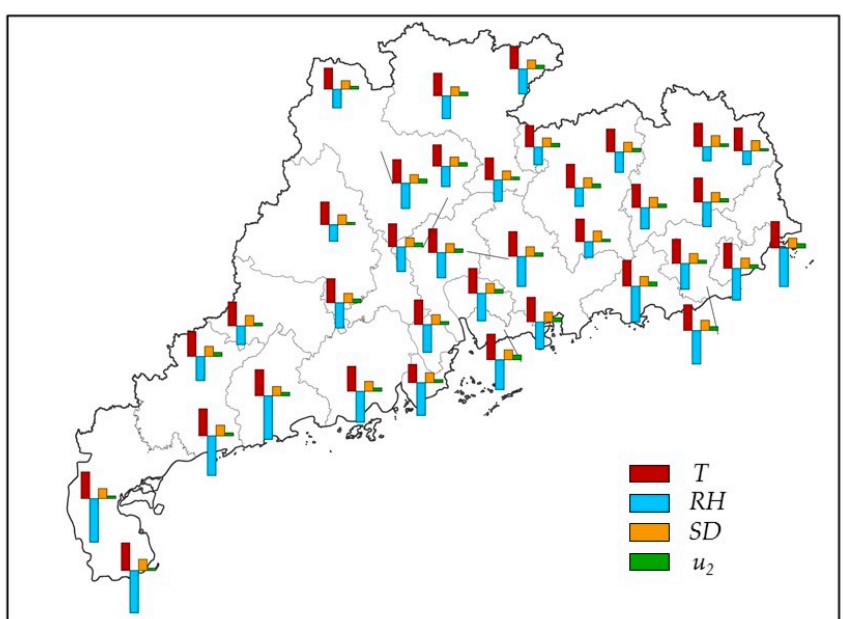

**Figure 5.** Spatial distribution of annual sensitivity coefficient of $ET_0$ to climatic factors in study area: temperature ($T$), relative humidity ($RH$), sunshine duration ($SD$), and wind speed ($u_2$).

On the seasonal scale (Figure 6), $ET_0$ showed positive $S_x$ for $T$, $SD$, and $u_2$ and negative $S_x$ for $RH$ in all seasons. The $S_x$ of $ET_0$ to $T$ was highest in summer and autumn, with an average of 0.73. The same measure was smaller in spring and winter, with values of 0.60 and 0.54, respectively. The $S_x$ of $ET_0$ to $RH$ varied significantly among different regions in all seasons, and the spatial distribution of $|S_x|$ increased from north to south. The $S_x$ to $RH$ was relatively high in winter and spring, with $|S_x|$ averages of 0.94 and 0.83, respectively, and smaller in autumn and summer, with $|S_x|$ averages of 0.63 and 0.55, respectively. The $S_x$ of $ET_0$ to $RH$ in different regions differed little between seasons, and the ranking of the $S_x$ to $SD$ was as follows: summer (0.35) > autumn (0.31) > spring (0.20) > winter (0.19). The $S_x$ for $u_2$ were ranked as follows: winter (0.15) > autumn (0.13) > spring (0.05) > summer (0.04). Overall, $ET_0$ was highly sensitive to $T$ and $RH$ in different seasons, while being sensitive to $SD$ in spring, summer, and autumn and moderately sensitive in winter. $ET_0$ was moderately sensitive to $u_2$ in winter, autumn, and spring, but negligibly sensitive in summer.

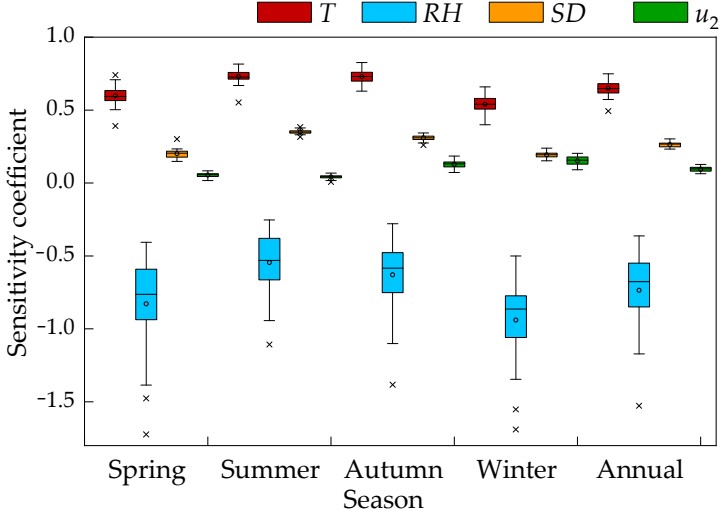

**Figure 6.** Box plots of sensitivity coefficient of $ET_0$ to climatic factors in study area: temperature ($T$), relative humidity ($RH$), sunshine duration ($SD$), and wind speed ($u_2$).

*3.4. Contributions of Climatic Factors to the Trends in ET$_0$*

We calculated the contribution rates of $T$, $SD$, $RH$, and $u_2$ to $ET_0$ using Equation (10), before adding them to obtain the total contribution rate of climatic factor changes to $ET_0$, which were noted as $ET_{0\text{-estimated}}$. Next, the actual relative rate of change in $ET_0$ was calculated using Equation (11), which was noted as a $ET_{0\text{-actual}}$. A correlation analysis between $ET_{0\text{-estimated}}$ and $ET_{0\text{-actual}}$ for all stations showed that $ET_{0\text{-estimated}}$ was relatively close to $ET_{0\text{-actual}}$ (Figure 7). The fitting points were concentrated around the 1:1 line, and the $R^2$ values were greater than 0.90 in different seasons and annually. Therefore, it could be considered reliable to quantify $ET_0$ changes based on the contributions of $T$, $SD$, $RH$, and $u_2$.

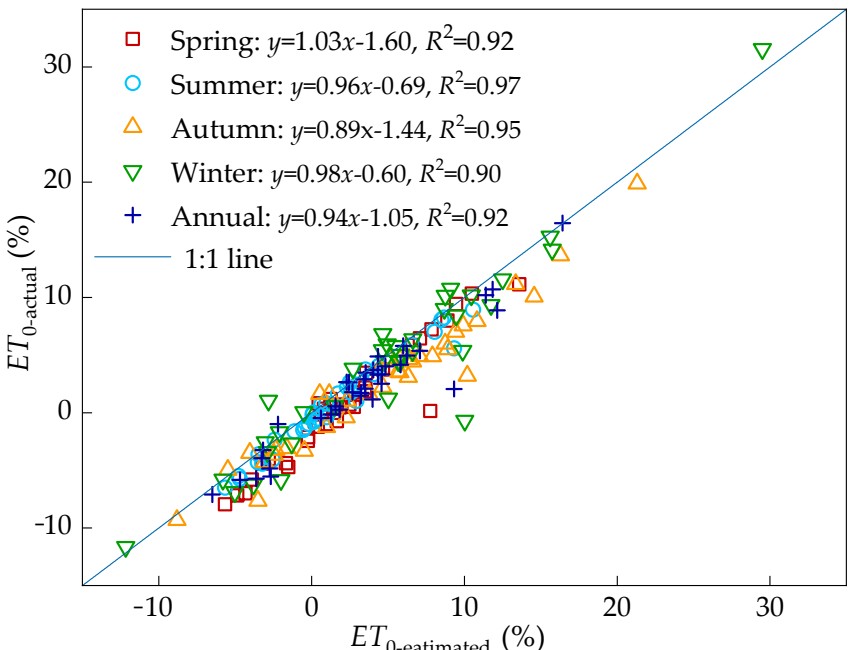

**Figure 7.** Relationship between estimated and actual relative variations in $ET_0$ in study area.

The spatial distribution of the contribution rate ($C_x$) of $T$, $RH$, $SD$, and $u_2$ to $ET_0$ variation in the study area is shown in Figure 8. The results indicate that the $C_x$ of $T$ to $ET_0$ variation is positive at all stations, with an average of 3.78% and a range in variation from 0.96% to 8.16% (Table 5), with high values occurring in the PRD and eastern coastal areas, while $C_x$ is relatively small in the north and west. The $C_x$ of $RH$ to $ET_0$ variation ranged from −2.55% to 10.22%, with 31 stations having positive $C_x$ and 6 stations having negative $C_x$. Negative values were found in stations such as Zhanjiang, Shaoguan, and Nan'ao, where $RH$ showed an increasing trend during the study period. High values of $C_x$ were found in the PRD, and low values were mainly found in the north and west. The $C_x$ of $SD$ to $ET_0$ change ranged from −7.67% to 1.84%. $|C_x|$ high values were mainly found in the PRD, Luoding, and Heyuan. $C_x$ was negative in most regions and positive only in Huilai and Yingde, where $C_x$ was 0.90% and 1.84%, respectively. $SD$ showed an increasing trend in these two regions. The $C_x$ of $u_2$ to $ET_0$ variation at different stations ranged from −5.70% to 10.91%, with 21 stations having positive $C_x$ and 16 stations having negative $C_x$. Negative values were mainly found in coastal areas where $u_2$ decreased; however, in the remaining 21 stations, $u_2$ showed an increasing trend, resulting in an overall positive $C_x$ on average.

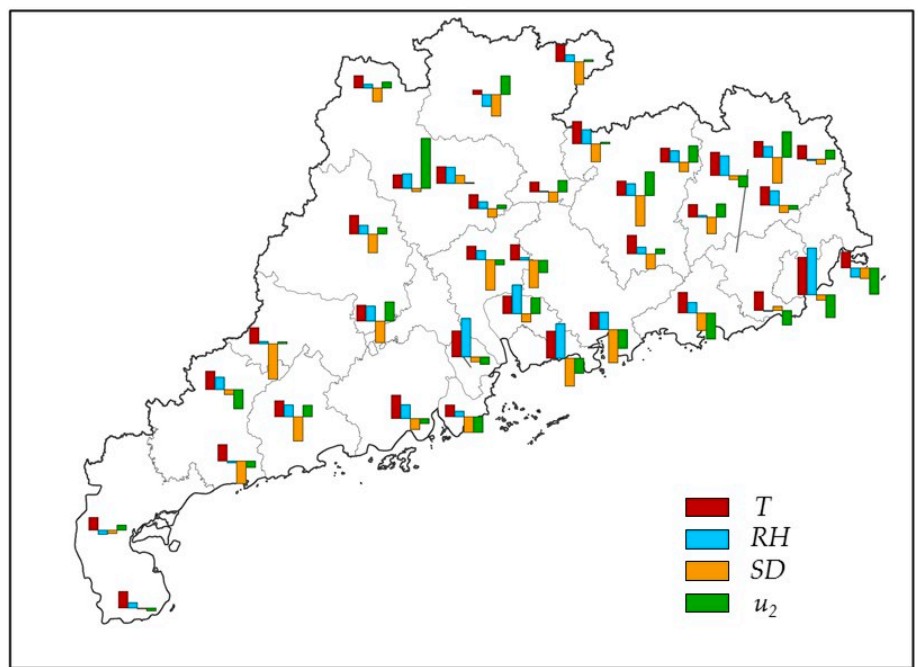

**Figure 8.** Spatial distribution of contribution rate of meteorological factors to $ET_0$ variations: temperature ($T$), relative humidity ($RH$), sunshine duration ($SD$), and wind speed ($u_2$).

**Table 5.** Relative changes in climate variables for different seasons and their contributions to $ET_0$ change in study area.

| Season | Relative Change $R_c$ (%) | | | | | Sensitivity Coefficient $S_x$ | | | | Contribution Rate $C_x$ (%) | | | |
|---|---|---|---|---|---|---|---|---|---|---|---|---|---|
| | $T$ | $RH$ | $SD$ | $u_2$ | $ET_0$ | $T$ | $RH$ | $SD$ | $u_2$ | $T$ | $RH$ | $SD$ | $u_2$ |
| Spring | 4.17 | −2.93 | −14.68 | 1.77 | −0.27 | 0.60 | −0.83 | 0.20 | 0.05 | 2.51 | 2.28 | −2.88 | 0.12 |
| Summer | 3.49 | −2.78 | −9.73 | 13.07 | 0.07 | 0.73 | −0.55 | 0.35 | 0.04 | 2.55 | 1.40 | −3.37 | 0.73 |
| Autumn | 6.11 | −4.61 | −9.26 | 3.95 | 2.15 | 0.73 | −0.63 | 0.31 | 0.13 | 4.47 | 2.61 | −2.86 | 0.22 |
| Winter | 10.40 | −3.15 | −17.58 | 0.92 | 3.22 | 0.54 | −0.94 | 0.19 | 0.15 | 5.59 | 2.67 | −3.39 | −0.22 |
| Annual | 6.04 | −3.37 | −12.81 | 4.93 | 1.29 | 0.65 | −0.73 | 0.26 | 0.09 | 3.78 | 2.24 | −3.12 | 0.21 |

Overall, the ranking of the contribution of each meteorological factor to $ET_0$ in the study area was $T$ (3.78%) > $SD$ (3.27%) > $u_2$ (2.73%) > $RH$ (2.58%). The $C_x$ of $T$, $RH$, and $u_2$ to $ET_0$ was positive on average, indicating that the temperature changes, $RH$, and $u_2$ in Guangdong over the last 61 years caused an increase in $ET_0$. In contrast, the $C_x$ of $SD$ to $ET_0$ was negative on average, indicating that the changes in $SD$ in Guangdong decreased $ET_0$ during the study period.

On the seasonal scale (Figure 9), the $C_x$ of $T$ to $ET_0$ change was positive in all seasons, with high $C_x$ mainly found in the PRD and the eastern coastal region. The mean $C_x$ gradually increased from spring (2.51%) to winter (5.59%) (Table 5), which is consistent with the ranking of $T$ tendency rate in different seasons. The $C_x$ of $RH$ to $ET_0$ variation was the smallest in summer (1.40%) and the highest in winter (2.67%), mainly due to the negative tendency rate and negative $S_x$ of $RH$ in different seasons. High $C_x$ was observed in the PRD in all seasons. The average $C_x$ of $RH$ to $ET_0$ changes in all seasons was negative, with the highest $|S_x|$ observed in winter (3.39%), followed by summer (3.37%), while the lowest was observed in autumn (2.86%). High $|S_x|$ values were mainly found in the PRD, Luoding, and Heyuan. The high value area of $|S_x|$ in winter was relatively lower than in other seasons. In conclusion, the contribution rate of $T$ and $SD$ to $ET_0$ change was higher in different seasons, followed by $RH$, while $u_2$ was very small. The $C_x$ of $SD$ was higher than $T$ in spring and summer and was the dominant factor influencing $ET_0$ variation. However,

in autumn and winter, the $C_x$ of $T$ was higher than *SD*, and $T$ became the dominant factor of $ET_0$ change in the study area.

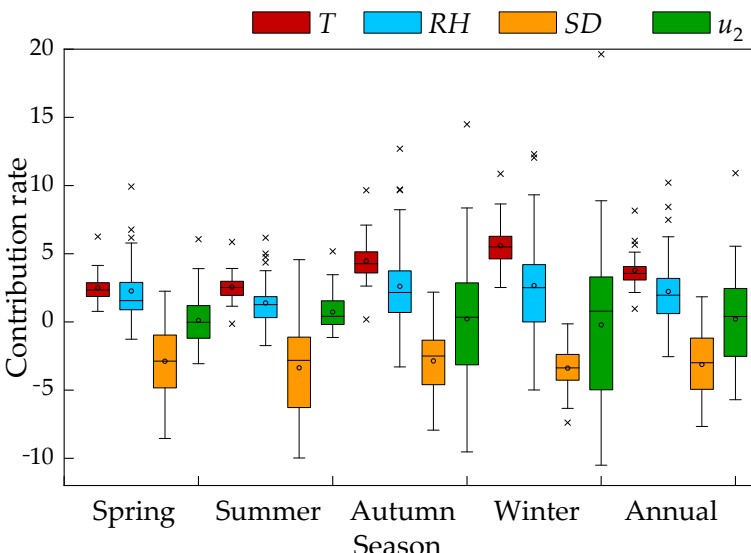

**Figure 9.** Contribution rate of meteorological factors to annual and seasonal $ET_0$: temperature ($T$), relative humidity (*RH*), sunshine duration (*SD*), and wind speed ($u_2$).

## 4. Discussion

### 4.1. Changing Trends of Meteorological Factors and $ET_0$

Over the past few decades, the air temperatures in most regions showed an unprecedented increasing trend, making global warming is indisputable. The overall $T$ in Guangdong increased at a rate of 0.17 °C/10a during the study period, and the main reason explaining the regional warming is the increase in greenhouse gas emissions due to global population growth and economic development. The decreasing trend of *RH* in the study area and the increase in $T$ indicate that the climate became drier to some extent over the 61 years studied. *SD* in Guangdong is on a downward trend; in fact, most of the world, such as Asia and Europe, is experiencing a decrease in sunshine hours to varying degrees, i.e., global dimming [38–40]. It was previously reported that human activities can cause a reduction in *SD* because atmospheric pollution from human activities leads to an increase in aerosols in the air. The aerosols increase the reflection and absorption of sunlight by the atmosphere, which, in turn, reduces the solar radiation reaching the ground, causing the reduction in *SD* [41]. Moreover, the decrease in wind speed makes the pollutants in the atmosphere less diffusible, which increases the near-surface aerosol concentration and contributes to the decrease in *SD* to some extent. $u_2$ in the study area decreases along with the trend in global wind speeds [42]. In this study, the distribution of $u_2$ changes was irregular, with stronger declines in Nan'ao, Shanwei, Shantou, and Shenzhen; these declines were concentrated in the central–eastern coastal region. Changes in meteorological factors also showed different seasonal characteristics variation. For example, $T$ increased in all seasons, and *RH* and *SD* decreased in different seasons. Meanwhile, $u_2$ increased only in summer and decreased in other seasons.

Annual $ET_0$ in Guangdong is slightly increasing at a rate of 2.76 mm/10a; it appears that many regions around the globe significantly influenced by oceanic climate recorded an increasing trend in $ET_0$, such as the Korean Peninsula, which is in a subtropical monsoon climate zone [43]; Paraíba, Brazil, and Madagascar, which are in tropical high-temperature climate zones [44]; and Austria, which is in a maritime temperate broad-leaved forest climate zone [45]. This trend is different from those recorded in other regions in China, such as the Yellow River Basin [46], North China Plain [47], Northwest China [36], and Beijing–Tianjin–Hebei regions [48], where $ET_0$ decreased since the 1960s, creating an "evap-

oration paradox". This paradox exists at about 62% of the stations across China, where $ET_0$ decreases at a rate of 5.2 mm/10a despite an increase in temperature [14]. However, this study showed some subtle differences in the spatial distribution in Guangdong: due to the influence of topography and geomorphology, $ET_0$ is higher in the southern low-elevation coastal areas than in the northern high-elevation areas. The evaporation paradox phenomenon was found at 12 of 37 stations in the study area, where $ET_0$ decreased with increasing temperature. Among these stations, Luoding and Dianbai in western Guangdong, Guangzhou, Zhuhai, Zengcheng, and Huiyang in the PRD, as well as 6 stations in Nan'ao in eastern Guangdong, showed significant decreasing trends in $ET_0$. The study also revealed seasonal variations in $ET_0$, with a decreasing trend in spring and an increasing trend in the other seasons, particularly autumn and winter. These results show that various factors affect $ET_0$, with each factor having a different weight.

*4.2. Climatic Factors Affecting the Variation in $ET_0$*

Climate change is the key factor driving $ET_0$ variation; however, there are differences and uncertainties in the factors influencing $ET_0$ variation between global regions, mainly due to the interactions between meteorological elements. It is generally accepted that $ET_0$ is positively correlated with $u_2$, $T$, and $SD$ and negatively correlated with $RH$ [49]. The sensitivity analysis conducted in this study shows that $ET_0$ in Guangdong is highly sensitive to $RH$, $T$, and $SD$ and moderately sensitive to $u_2$, which is consistent with the results recorded for the Poyang Lake catchment, China [32], and for the Korean Peninsula [43]. However, the sensitivity of $ET_0$ to climatic factors varies across different regions, and $ET_0$ was most sensitive to $RH$ in Guizhou Province [50], Jiangsu Province [23], the Beijing–Tianjin–Hebei region [48], and the Huai River Basin [51]. In contrast, $ET_0$ was most sensitive to $u_2$ in the northwest inland region, followed by $RH$, $T$, and $SD$ [36]. Although $ET_0$ was more sensitive to $RH$ than to $T$, the contribution analysis showed that $RH$ contributed less to the increase in $ET_0$ than $T$ and $SD$. In the tropical high-temperature climate zone in Brazil, the main climatic factor driving $ET_0$ changes is temperature, while the most critical impact factor alternates between temperature and sunshine hours in the rainy and dry seasons. In Austria, which is located in a temperate broad-leaved forest climate zone, the main reason for $ET_0$ rise is the increase in solar radiation [45]. In this study, $T$ was the main cause of $ET_0$ changes in Guangdong, and $SD$ was the second main cause of $ET_0$ changes, while in spring and summer, $SD$ was the dominant factor of $ET_0$ changes because $SD$ decreased more significantly. It was reported that the drier the climate in China, the greater the contribution of wind speed to $ET_0$, especially in the arid northwest, where $u_2$ is the main cause of $ET_0$ decrease [26,36]. The results of this study showed a relatively large contribution of $u_2$ to $ET_0$ variation in summer, and the same low $RH$ and strong $u_2$ rise were found at these stations.

Overall, the increasing effect of rising $T$ and falling $RH$ on $ET_0$ in Guangdong during the study period exceeded the decreasing effect of falling $SD$ on $ET_0$, ultimately leading to an overall increasing trend of $ET_0$. However, in regions where the evaporation paradox exists, i.e., $T$ rise is accompanied by $ET_0$ decline, the contribution of $SD$ to $ET_0$ is usually more significant than that of $T$. Moreover, in inland areas of Guangdong, such as Guangzhou, Zengcheng, Huiyang, and Luoding, the strong decreasing effect of $SD$ on $ET_0$ masks the increasing effect of $T$ and $RH$, while in coastal regions, such as Dianbai and Nan'ao, the decrease in $SD$ and rise in $RH$ offset the increase in $ET_0$, suggesting that there are spatial differences and uncertainties in the weights of each factor affecting the variation in $ET_0$.

*4.3. Impact of Climate and $ET_0$ Changes on Agricultural Production*

Guangdong belongs to the tropical and subtropical monsoon climate zone, with a humid climate, abundant heat, and abundant but unevenly distributed precipitation. This region is a significant producer of grain crops (e.g., rice, corn, and tubers) and tropical crops (e.g., sugarcane, rubber trees) in China [52,53]. Climate change may complexly impact agricultural production and water resource management in this region. In fact,

an increasing drought trend was consistently observed in recent years, which was both global and regional in scale, including in the Guangdong [54,55]. Climate warming will result in a richer agro-climatic heat resource, a longer crop growing season, and more heat in the growing season in Guangdong. This change, in turn, will push the existing agro-climatic zone and crop maturity boundaries northward and to higher elevations, which favors the cultivation of tropical crops in the region, while the northern boundary of the second and third maturity zones of crops also moves northward and the area is expanded. However, the increase in temperature may also lead to drought and summer heat disasters, which could reduce agricultural yield or affect the quality of crops. The decrease in sunshine duration will negatively impact the high and stable yield factors of tropical fruits in the region, such as fruit enlargement and sugar accumulation in sugarcane, as well as increase the late rice seed setting rate [53]. In addition, $ET_0$ increased in the province, particularly in eastern Guangdong and the Leizhou Peninsula. Crops' evapotranspiration water consumption in these agricultural areas increased, leading to an increase in demand for irrigation water. The contradiction between the supply of and demand for water resources in the future is prominent, which may aggravate the drought and water shortage situation in water-restricted areas, especially in spring and winter, when precipitation is at the lowest annual level. Corrective and preventive measures should be considered in water resources planning and agricultural production.

For a complete understanding of the mechanism driving changes in regional evapotranspiration responses to climate change, further attention should be paid to the feedback and quantitative relationship between actual evapotranspiration and $ET_0$, as well as how this relationship affects regional hydrological cycles.

## 5. Conclusions

In this study, we conducted a comprehensive analysis of the trends of $ET_0$ and major climatic factors (*T*, *RH*, *SD*, and $u_2$) in Guangdong from 1960 to 2020. Our findings provide important insights into the factors driving variations in $ET_0$ and implications for future regional water management. Based on the results, the following conclusions can be made:

(1) The annual $ET_0$ in Guangdong increased at a rate of 1.61 mm/10a. $ET_0$ increased at most stations; only 6 stations had a decreasing trend for all 37 analyzed samples. $ET_0$ decreased in spring and increased in the other seasons, though these trends were not statistically significant. Meanwhile, *T* significantly increased during the study period, while *RH* and *SD* significantly decreased.

(2) Sensitivity analysis showed that $ET_0$ was more sensitive to *RH* and *T* than *SD* and $u_2$ in Guangdong. $ET_0$ was most sensitive to *RH* in spring and winter and *T* in summer and autumn. Through considering the variation in variables and their sensitivity to $ET_0$, the results showed that *T* was the dominant factor for $ET_0$ variation in Guangdong, followed by *SD*. In the areas where the "evaporation paradox" occurs, as well as in spring and summer, *SD* was the dominant factor in $ET_0$ variation. Therefore, the trend of climatic factors plays a critical role in analyzing the variation in $ET_0$.

(3) The increasing effect of rising *T* and decreasing *RH* on $ET_0$ masked the decreasing effect of *SD* on $ET_0$, resulting in an overall increase in $ET_0$ in Guangdong. This result suggests that a potential future increase in *SD* combined with a decrease in *RH* may lead to higher evapotranspiration rates and drought events in Guangdong. Therefore, we recommend adopting long-term water management strategies for sustainable development to cope with regional climate change.

**Author Contributions:** Conceptualization, Formal analysis, B.Z. and D.A.; Data Curation, B.Z. and C.Y.; Validation, R.K.; supervision, J.S.; writing—original draft preparation, B.Z.; writing—review and editing, H.Y. project administration, J.S. All authors have read and agreed to the published version of the manuscript.

**Funding:** This research was funded by the Hainan Provincial Natural Science Foundation of China (grant number 322QN415), and the Central Public-Interest Scientific Institution Basal Research Fund (grant number 1630062023008, 1630102022002, 1630102022004).

**Data Availability Statement:** The data presented in this study are available upon request from the corresponding authors.

**Acknowledgments:** The authors would like to thank the editors and anonymous reviewers for their constructive comments and suggestions that helped us to improve the quality of this manuscript.

**Conflicts of Interest:** The authors declare no conflict of interest.

## Nomenclature

| | |
|---|---|
| $\Delta$ | Slope of the vapor pressure curve (kPa $°C^{-1}$) |
| $\gamma$ | Psychrometric constant (kPa $°C^{-1}$). |
| $\beta$ | Climatic tendency rate |
| $a$ | Linear slope |
| $b$ | Intercept |
| $C_x$ | Contribution rate (%) |
| $e_a$ | Actual vapor pressure (kPa) |
| $e_s$ | Saturation vapor pressure (kPa) |
| FAO56 PM method | FAO56 Penman–Monteith method |
| PRD | Pearl River Delta |
| $ET$ | Evapotranspiration (mm $d^{-1}$) |
| $ET_0$ | Reference evapotranspiration (mm $d^{-1}$) |
| $G$ | Soil heat flux (MJ $m^{-2}$ $d^{-1}$) |
| $n$ | Length of the data set |
| $p$ | Significance test value |
| $R_c$ | Relative change rate of certain meteorological factors (%) |
| $RH$ | Relative humidity |
| $R_n$ | Net radiation (MJ $m^{-2}$ $d^{-1}$) |
| $S$ | Test statistic |
| $SD$ | Sunshine duration (h) |
| $S_x$ | Sensitivity coefficient |
| $T$ | Air temperature (°C) |
| $T_{max}$ | Maximum temperature (°C) |
| $T_{min}$ | Minimum temperature (°C) |
| $u_2$ | Wind speed at 2 m (m $s^{-1}$) |
| $u_z$ | Wind speed at $z$ m (m $s^{-1}$) |
| $x$ | Meteorological factors |
| $\overline{x}$ | Mean of the meteorological factor time series. |
| $Z$ | Standardized test statistic |

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
