# Peer review of "Spatiotemporal Variations of Reference Evapotranspiration and Its Climatic Driving Factors in Guangdong, a Humid Subtropical Province of South China"

_agronomy, doi:10.3390/agronomy13061446_

Round 1

Reviewer 1 Report

Dear Authors,

I have had the opportunity to read your well-edited MS. It is commendable that the logical thread was retained throughout the entire MS.

The numerous abbreviations could be presented in a list at the beginning of the MS.

The discussion contains relatively few citations. It is expressly true for sub-ch. 4.3.

For the study not to remain at the regional level and interest, it would be worthwhile to refer more to global environmental changes.

What are the most important outcomes in the global sense? For example, regions with similar climates should be considered. I suggest using the Köppen-Geiger classification system to find similar regions.

It could also explain better the Introduction and the Discussion.

It would be interesting to the readership to explain how climatic changes can alter the relevant Sustainable Development Goals in the future in the studied region based on the experiences derived from the recent climatic data.

Sincerely,

the reviewer

Minor grammatic errors were found. Please, check again the MS in this sense.

Reviewer 2 Report

Dear,

After reading the work, I consider that it is of great importance for the knowledge of the area and that the information obtained improves current knowledge. I consider that the work can be published with a few suggestions present in the attached file.

Sincerely,

Reviewer 3 Report

The manuscript is well-written and describes findings of interest to water managers.  While results are regional, the methodology and findings have relevance to researchers in other regions. 

My comments are in the attached annotated file.  My only major recommendations is that Figure 7 and the discussion of it could be deleted.  If retained, the manuscript needs to be revised to make it clear why this figure is relevant to the paper.  I did not find the ETo-estimated to be clear or relevant. 
